# Designing Spatial Visualization and Interactions of Immersive Sankey Diagram in Virtual Reality

## ABSTRACT

Virtual reality (VR) is a revolutionary method of presenting data visualizations, which brings potential possibilities for enhancing analytical activities. However, applying this method to visualize complex data flows remains largely underexplored, especially the Sankey diagrams, which have an advantageous capacity to represent trends in data flows. In this work, we explored a novel design for the immersive Sankey diagram system within VR environments, utilizing a three-dimensional visual design and several interaction techniques that leveraged VR's spatial and immersive capabilities. Through two comparative user studies, we found the effectiveness of the VR Sankey diagram system in improving task performance and engagement and reducing cognitive workload in complex data analysis. We contribute to an interactive, immersive Sankey diagram system in VR environments, empirical evidence of its advantages, and design lessons for future immersive visualization tools.

## CCS CONCEPTS

• **Human-centered computing** → **Visualization design and evaluation methods**; **Virtual reality**; *User studies.*

## KEYWORDS

data visualization, virtual reality, immersive analytics, sankey diagram

## 1 INTRODUCTION

Data visualization is a powerful method to help people understand data and communicate insights by converting information into graphics or images [20, 40]. Especially in the era of data and information explosion, visualization technology has revolutionized how we present and decipher complex datasets. Meanwhile, the advent of immersive technologies such as Virtual Reality (VR) fosters a new research area in visualization – Immersive Analytics (IA) [10]. The inherent characteristics of VR, such as infinite space and immersion, make data visualization free from two-dimensional screen limitations. As some data analysts stated, VR technology is a potent medium to offer a wide field of vision, augmented dimensions, and a sense of presence, thereby enhancing the analysis of scientific data [25, 41].

While the potential of Immersive Analytics has been demonstrated, some specific applications, such as visualizing data flows in VR environments using Sankey diagrams, have not been well explored. The Sankey diagram stands out for its capacity to visualize data flows across diverse fields [15, 35], i.e., economy [33] and mechanical engineering [30]. It enables non-professional users to quickly understand the relationships and connections between data points in a visually accessible and interpretable way [28]. As far as we know, prior research has not explored how to integrate spatial visualizations and interactions into conventional Sankey diagrams, which leaves a gap in developing such a practical system in VR.

Based on such observations, we proposed a novel 3D Sankey diagram design with a staggered and spatially distributed layout to enhance its visual clarity. Compared to the traditional 2D form of the Sankey diagram which migrated directly from the desktop-based system into VR, we conducted a user study that identifies the 3D Sankey diagrams' benefits in displaying intricate data flows, thus enhancing comprehension of data flows. Based on these findings, we finalized a VR Sankey diagram system that further incorporates interactive features into the above Sankey diagram designs, leveraging VR's immersive and spatial capabilities. Through a comparative user study between our system and a desktop-based Sankey diagram system, the VR Sankey diagram system showed the advantages of better task performance, higher user engagement, and lower workload. Overall, we found that designing visualizations incorporating spatial depth cues and having interactions leveraging spatial capability in VR can intrigue a deeper comprehension and a better analytical experience in data exploration.

To summarize, our contributions are threefold. First, we developed a VR Sankey diagram system with a 3D visualization design that utilized spatial depth cues and interactive features that leveraged spatial capability in VR. Second, we conducted two empirical studies to demonstrate its advantages of presenting intricate data structures, enhancing comprehension, and offering a more engaged analytical experience over the desktop-based Sankey diagram system. Third, we contributed insights and design lessons through the two empirical studies, which also benefit designers who work in the broad field of future immersive data visualization systems.

## 2 RELATED WORK

### 2.1 Immersive Analytics

Immersive Analytics (IA) aims to improve data understanding and workflows and support decision-making by integrating data visualization with an immersive user interface and captivating embodied analysis tools [9, 14, 34]. Recently, IA has been applied in diverse fields such as healthcare [36], aviation [43], archaeology [23], geography [26], etc. Some researchers confirmed the benefits of transforming the information visualizations into 3D representations within the immersive space [29, 44, 45]. For instance, Millais et al. [29] and Filho et al. [45] have demonstrated the advantages of

**Figure 1: This figure presents the 3D and 2D Sankey diagrams featuring a high-density dataset in VR environments. (a) displays the 3D Sankey diagram from two perspectives, where shadows help clarify the nodes' hierarchical information. (b) illustrates the corresponding 2D Sankey diagram without considering VR's additional spatial depth dimension.**

presenting scatterplots and parallel planes in VR environments over traditional 2D data visualizations in accuracy and engagement.

Various tools have been developed to help users present information visualizations in immersive environments, such as ImAxes [7], IATK [6], 3D radar charts [39], and immersive Space-Time Cube [46]. These studies mainly focus on common visualizations such as scatterplots, node-link graphs, and geo-map visualizations. However, tools like Sankey diagrams for visualizing complex data flows have not been well explored [21].

Tadeja et al.'s work [42] is the most similar to ours. They explored the 3D adaption of Parallel coordinate plots in VR and studied VR PCP's potential benefits in data processing and comprehension. Inspired by their study, we proposed novel spatial visualizations and interactions of Sankey diagrams to make them suitable for immersive environments and developed an interactive system for an empirical study, aiming to get insights on whether it can bring unique benefits to analytical activities for data flows.

## 2.2 Sankey Diagram

Sankey diagrams typically depict a predetermined collection of layers (vertical lines) corresponding to various time intervals [48]. They keep evolving and are applied in a range of fields, such as the visualization of technical debt [33], product life cycles [8], and sequences of events [5], which facilitate a deeper understanding of complex systems for non-professionals [28]. The research on Sankey diagrams and their variants has become a pivotal area within the visualization domain, with researchers dedicating efforts to enhance the diagrams' visual effect [47, 48] to enable them to convey more information [1, 35], and to study their practical applications [27, 28]. However, previous work mainly focused on planar Sankey diagrams for desktop applications, with a limited exploration of 3D representations. Alemasoom et al. [1] proposed a 3D form by combining bar charts with the Sankey diagram in 3D space, in which bar charts are perpendicular to the 2D planar Sankey diagram, utilizing rotating views to show the additional information attached to each node. Neugebauer et al. [30] explored 3D Sankey diagrams for the development of energy-efficient products in mechanical engineering.

In contrast, our work specifically focuses on integrating spatial cues into Sankey diagrams without merging them with other chart types or simulation models. We proposed a novel 3D representation utilizing the depth cues within VR for visualizing Sankey diagrams

and a set of interactive features for 3D data exploration. Results show that our system brings better task performance compared to a conventional desktop-based Sankey diagram system without additional workload.

## 3 EVALUATION TASKS FOR SANKEY DIAGRAMS

Sankey diagrams are sophisticated flowcharts designed to depict flows from one set of values to another [11]. These diagrams consist of nodes that symbolize different categories and links that illustrate the data flows. The height of a node correlates with its value, making the data point value proportional to the node size. The breadth of a link correlates with its flow value, making the flow volume directly proportional to the link's width [47]. In our comprehensive review of research on Sankey diagrams, we selected seven tasks to evaluate Sankey diagrams. These tasks encompass detailed inspection of the diagrams' nodes, links, and directional flows and foundational analytical exercises, which evaluate the diagrams' clarity and effectiveness in enhancing comprehension.

**(1) Node Filtering (NF)** aims at searching for specific nodes under set criteria, similar to the "Filter" task introduced by Lee et al. [22] and Quadri et al. [37], and the "Find Element" task delineated by Gutwin et al. [15]. It is mainly used to check if the node is easy to observe.

**(2) Link Identification (LI)** identifies the presence or absence of specific links within the diagram, mirroring the "Object Identification" task posited by Mathis et al. [28] and the "Existence" task by Gutwin et al. [15]. It is mainly used to check if the link is easy to observe.

**(3) Retrieve Value (RV)** derived from Mathis et al.'s [28] work, checks out the value of a specific link to check the readability of hover labels.

**(4) Link Comparison (LC)** selects and compares two links by specific standards, mirroring a variant of Amar's "Filter" task [3]. It focuses on whether the size ratio is intuitive, combined with the RV task in some cases.

**(5) Extremum Identification (EI)** derived from Amar's "Find Extremum" task [3], identifies the links representing maximum and minimum values. It examines both the observation of the widest and finest link, combined with the RV task in some cases.

**(6) Count Links (CL)** focuses on determining the number of links flowing into or out of a particular node, the same as the "Count

Links" task proposed by Gutwin et al. [15]. It is aimed to evaluate links' clarity as they flow into or out of the node.

**(7) Path Analysis (PA)** stemming from the research endeavors of Lee et al. [22] and Mathis et al. [28], involves the determination of potential pathways between nodes across different layers, typically tracing from an origin node to the end node. It assesses the diagram's effectiveness in helping users do simple analysis.

We used the NF, LI, LC, EI, and CL tasks in the first study for visual clarity evaluation and all seven tasks in the second study for a comprehensive comparison. Specifically, we represent the tasks with a set of questions that participants need to answer to evaluate the task performance using the systems.

## 4 USER STUDY 1: 2D VS. 3D SANKEY DIAGRAMS IN VR

In this section, we conducted a user study to determine if introducing the VR's additional spatial depth into Sankey diagram designs can offer more visual advantages in clarifying the relationships between data points and the trends of links to enhance understanding of data flows in the Sankey diagrams (Figure 1). We first proposed a novel 3D Sankey diagram design that considers VR's unique spatial advantages and a 2D Sankey diagram design transferred directly from the desktop environment to the VR, as outlined in subsection 4.1. We then detailed the setup, procedures, and results of the comparative study of the 2D and 3D Sankey diagram designs.

### 4.1 2D & 3D Sankey Diagram Designs

The 2D Sankey diagram design in VR environments is a direct adaptation from the desktop-based Sankey diagram to VR environments without leveraging the additional spatial dimension inherent to VR. Specifically, the nodes and links of these Sankey diagrams are optimized by Integer Linear Programming (ILP), a method widely recognized for reducing link overlaps in Sankey diagrams. Figure 1 (b) provides an illustrative example of a 2D Sankey diagram in our user study.

Based on the 2D Sankey diagrams, we developed 3D Sankey diagrams by incorporating the additional spatial dimension of VR (i.e., depth) into the original 2D designs. Specifically, this enhancement involved expanding the thickness of each node within the diagram to facilitate users' understanding and remembering [18]. Furthermore, adhering to the perceptual principle whereby objects appear larger when closer and smaller when further away [13], we rearranged the nodes within each layer based on their size, utilizing VR's additional spatial depth dimension. This arrangement positions larger nodes at the forefront and smaller ones at the rear. It was adopted for two main reasons. Firstly, it aids in transitioning the links between nodes from a 2D plane to a 3D space, creating a staggered and spatially distributed layout. This aims to reduce visual overlap and enhance the distinction between links. Secondly, it further draws user attention to larger nodes, underscoring the principle that the size of visual elements signifies their importance [24]. This is designed to streamline user efforts by directing focus towards important elements. To further clarify the hierarchical information of the nodes, we added a shadow to each node. As shown in Figure 1 (a), the 3D Sankey diagram design corresponding to the 2D design depicted in Figure 1 (b), illustrating

how the 3D design leverages the additional spatial depth dimension to create a different layout.

Note that we did not consider any interactive features within the diagrams since our goal was to assess the visual effectiveness of the new diagram designs specifically. However, we retained the fundamental functionalities of VR, i.e., users can still navigate and adjust their viewpoint to facilitate a comprehensive evaluation.

### 4.2 Study Setup

*4.2.1 Participants.* We recruited 7 participants with normal vision without visual impairments (4 males and 3 females aged 25 ± 4.14). Within this group, two individuals had professional experience in data visualization. While one participant was unfamiliar with Sankey diagrams, four had heard of or somewhat knew about the concept, and two were familiar with and had previously used Sankey diagrams. Regarding VR experience, one participant had none, whereas the remaining six had limited exposure to VR technology, having used VR devices fewer than 10 times.

*4.2.2 Materials.* We selected 5 tasks (NF, LI, LC, EI, and CL) to compare the visual performance between 2D and 3D Sankey diagrams since these tasks mainly focus on the inspection of the visual features of the diagram (nodes and links). We generated four datasets by controlling the number of nodes, links, and layers to regulate data density: two were designated as low-density and two as high-density. The Sankey diagram representing low-density data featured 10 nodes and 25 links, spanned 3 layers. The high-density data diagram comprised 30 nodes and 75 links, spanned 6 layers. We gave the data with real-world meanings to make the Sankey diagrams understandable. Each node is named after a country, and the links represent the monetary value of trade flows between them. We randomly generated a unique color for each node from a color wheel and matched the color of each link with its source node. This color scheme was consistently applied across both 2D and 3D Sankey diagrams. For each dataset, we created 5 questions encompassing these 5 tasks. The questions can be answered by observing the diagrams without additional information.

*4.2.3 Tasks.* In this study, participants were required to inspect and understand data flows in four Sankey diagrams within VR, i.e., two 2D Sankey diagrams with low-density and high-density datasets and two 3D Sankey diagrams with low-density and high-density datasets. During the inspection of each diagram, participants responded to five questions tailored to the respective dataset, completing 20 questions in total. The process was structured to begin with the low-density diagrams before progressing to those of high-density. To mitigate potential inter-individual differences or learning effects, the sequence in which the 2D and 3D diagrams were presented was counterbalanced and randomized for each participant.

*4.2.4 Procedure.* The study lasted approximately 30 minutes, with all sessions conducted in face-to-face, one-on-one meetings. Participants were asked to provide demographic information before the experiment. Then, we introduced them to the fundamental concepts of Sankey diagrams and the operations of navigation and view adjustment in VR. The experiment began after they were familiar with the operations and VR environment. The questions would appear at

the bottom of the user's field of view in VR. Participants answered questions verbally. Following each question, they evaluated the task's ease via a Single Ease Question (SEQ) within the VR context before moving on to the next question. After completing two series of questions related to two low-density datasets, we provided participants with a five-minute break to mitigate fatigue and ensure participants' physical readiness for the subsequent experiments.

*4.2.5 Implementation.* The VR environment is built on Unity 3D with Oculus Quest3 and Oculus Touch Controller with a smooth frame rate that exceeds 80 FPS.

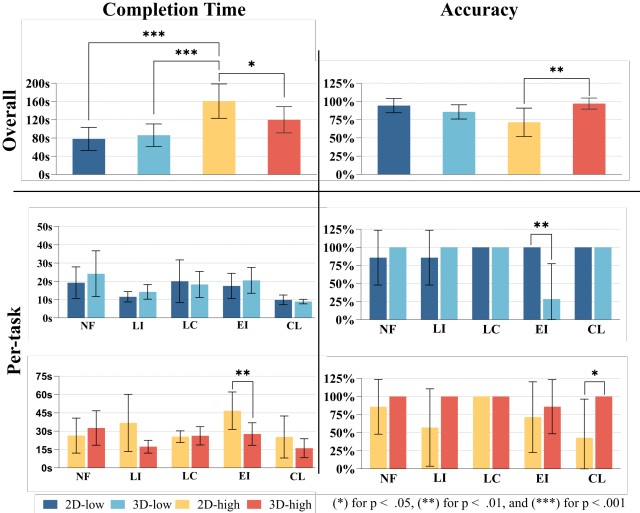

**Figure 2: Averages and standard deviations for task completion times and accuracy rates.**

## 4.3 Result Analysis

In this section, we report the quantitative results and user feedback.

*4.3.1 Task Performance.* We adopted several metrics to evaluate the task performance, including the completion time, task accuracy, and SEQ scores. The SEQ score is immediate user feedback on task difficulty, assessing the perceived ease of completing the task.

Figure 2 presents averages and standard deviations for completion time and accuracy of overall and per-task in the study. Participants had a better understanding of the Sankey diagrams with low-density data compared with those with high-density data, demonstrated by a considerably less overall task completion time (p < .001), slightly higher task accuracy and SEQ scores (p > .05). Notably, in high-density datasets, participants understood the 3D Sankey diagram much better than 2D with less overall completion time (p < .05), higher accuracy (p < .01) and higher SEQ scores in the LI and EI tasks (p < .001).

The factor of data density (p < .001) presented a significant influence on the overall task completion time, while the diagram form factor did not show any significance (p > .05). To figure out the impact of diagram forms on task completion time, we employed an ANOVA test on each data density. Within low-density datasets, the forms do not significantly impact the overall and per-task completion time. Conversely, in high-density datasets, participants had

a quicker understanding of the 3D Sankey diagram than the 2D, indicated by less completion time of the overall tasks (p < .05) and the LC task (p < .05).

We used the Wilcoxon rank-sum test for accuracy analysis. In low-density datasets, the tasks marginally outperformed using the 2D Sankey diagram over the 3D (p > .05). Notably, in the EI tasks, participants had a more precise understanding of the 2D Sankey diagram than the 3D, reflected by the average accuracy rates (p < .01). In high-density datasets, participants more accurately understood the 3D Sankey diagram than the 2D, with significantly higher accuracy rates in overall tasks (p < .01) and the CL task (p < .05).

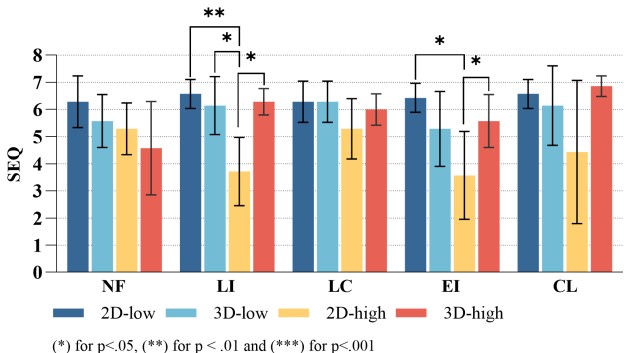

(*) for p<.05, (**) for p < .01 and (***) for p<.001

**Figure 3: Averages and standard deviations for the Single Ease Question scores (7 represents very easy).**

Analyzing responses to Single Ease Question (SEQ) presented immediately after each task, as shown in Figure 3. The results show that participants found it easier to use the 3D Sankey diagram to complete the LI (p < .01) and EI (p < .05) tasks in the high-density dataset. Also, they performed slightly better in the two tasks with higher accuracy and shorter completion time (p > .05). This indicates that the 3D Sankey diagram makes it easier to inspect links in high-density datasets.

*4.3.2 User Feedback.* Six of seven participants preferred 3D Sankey diagrams for their immersive and engaging observation experience, as they offered varied perspectives. One participant favored 2D diagrams, citing discomfort in navigating a VR environment and preferring static viewing. Some participants noted little difference in visual forms in low-density data scenarios, while some thought the 3D form made the diagram more complex. However, most participants commented that the 3D Sankey diagram had better visual clarity in the high-density data scenarios. One of them said, "*The 3D diagram reduces link overlap in complex data, and I can avoid occlusion by adjusting the position and viewing angle. However, the links of the 2D Sankey diagram overlap, leading to color blending and making link tracing harder.*" Besides, users offered some optimal suggestions like "*address fine link visibility*", "*enhance color contrast issues*", and "*add interactive features*".

To summarize, we found that 2D Sankey diagrams and 3D Sankey diagrams each have advantages in data understanding under different tasks within different data densities. The 3D Sankey diagrams leveraging depth cues could enhance the clarity of complex data structures and provide varied perspectives and a more immersive experience, thus enhancing the understanding of data flows. In

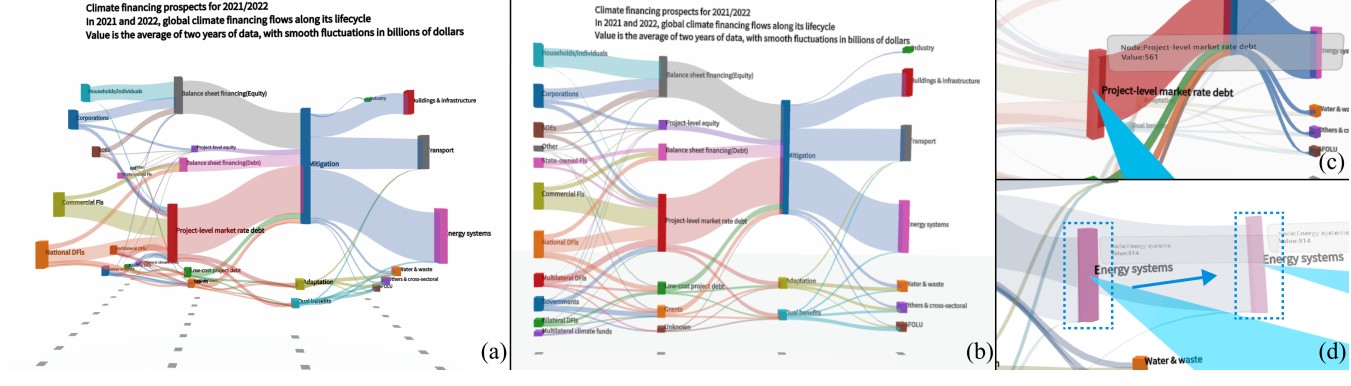

**Figure 4: The immersive Sankey diagram system in VR. (a) The 3D form of the system. (b) The 2D form of the system. (c) The spatial highlighting feature and the hover label feature. (d) The dragging feature.**

contrast, the 2D Sankey diagrams are more straightforward to interpret with simple data than the 3D. Therefore, we argue that it is essential to consider the benefits of both 2D and 3D forms before using them. Both forms may be required in VR, and offering users options to switch between them will trigger a better understanding of data flows.

## 5 SPATIAL INTERACTIONS

Based on the insights from the above study, we further detail the design of an immersive Sankey diagram system in a VR environment, with visual improvements and interactive features. For the visual improvements, we adjusted the color settings, adopted the D3's category 10 color palette, and reduced the opacity of narrow links to enhance clarity. We then proposed the following interaction techniques that could potentially augment 3D data exploration. All the interactions can be triggered by the Oculus Touch Controller.

*Movement & Rotation* Navigation is facilitated via the Thumbstick on the right hand for viewpoint rotation and on the left for adjusting altitude and moving around. Users can also move their heads to shift perspective.

*Hover labels* When the controller's ray targets a node or link from the right hand, its color changes from white to blue, indicating selection. This action also displays a label, revealing detailed information such as numbers about the selected node or link, as shown in Figure 4 (c).

*Spatial highlighting* Taking advantage of the evidence that stereoscopic "pop-out" effects significantly enhance the visibility of elements [2]. Our highlighting feature is intelligently designed to enhance the color and emphasize the spatial positioning of elements, as seen in Figure 4 (c). The feature is activated by clicking the Trigger button on the right-hand controller when the ray selects a node. We ensure that the highlighted elements will remain prominently displayed no matter how the user moves.

*Dragging* By utilizing the Grip Button, users can easily select and reposition nodes while preserving the links between the nodes that adapt to any changes in placement, as seen in Figure 4 (d).

*Form switching* Based on user study 1, we set this functionality to allow users to switch the Sankey diagram between 2D (Figure 4b) and 3D (Figure 4a) forms by pressing the X button on the left controller. In the 2D form, we kept the thickness of nodes to make it more harmonious in three-dimensional space.

*Reset* We provided two reset options: diagram reset (A Button) and position reset (B Button) to help users retrieve information. The diagram reset helps users move the nodes back to their original positions after highlighting or dragging operations, while the position reset reverts the users' spatial orientation to face the diagram.

## 6 USER STUDY 2: VR SANKEY DIAGRAM SYSTEM EVALUATION

We conducted a comparative analysis with a traditional desktop-based system (Figure 5) to evaluate the usability and effectiveness of our system (Figure 4). We aim to gain insights into the performance of our immersive system and a deeper understanding of how it would benefit analytical activities for data flows.

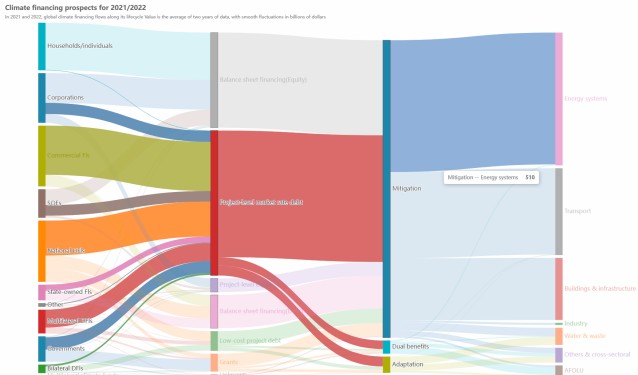

**Figure 5: The desktop Sankey diagram system.**

### 6.1 Study Setup

*6.1.1 Participants.* We recruited 16 students from our university campus (7 females, mean age = 24, SD = 2.04). They come from diverse backgrounds spanning 14 majors, such as energy power, optimal engineering, clinical medicine, law, etc. All participants had normal vision. Within this group, seven had professional experience in data visualization. Only two had heard of or known about Sankey Diagrams, and the rest of the participants were entirely unfamiliar with them. Regarding VR experience, one participant had

no experience with VR, one used VR devices frequently, two had 5 to 10 times VR exposure, and the remaining twelve had limited VR exposure (fewer than 5 times).

*6.1.2 Materials.* We selected two real-world datasets from different fields to judge the practicability of our system. One is about domestic energy flows in Canada in 2013 [38] (31 nodes, 65 links, and 7 layers), and the other one is about the global landscape of climate finance in 2021 and 2022 [17] (28 nodes, 69 links, and 4 layers). For each dataset, we created 7 questions encompassing these 7 tasks. The specifics of these questions are detailed in the supplementary material.

*6.1.3 Tasks.* Our tasks, i.e., a set of questions, aim to comprehensively evaluate the visual effects, interactive features, and the impact on users' comprehension of the two systems. Given that each set of questions is associated with a different dataset, participants were tasked with completing a total of 14 questions. To minimize learning effects, we employed a counterbalanced approach for the order in which the designs and datasets were presented, assigning them randomly to participants.

*6.1.4 Procedures.* The study lasted ~40 minutes by face-to-face, one-on-one meetings. Participants were asked to provide demographic information before the experiment started. Then, we introduced them to the fundamental concepts of Sankey diagrams and the operations of both the desktop and VR systems. The experiment began after they were familiar with the systems. In the VR system, questions would appear at the bottom of the user's field of view, while for the desktop system, questions appeared on the right side of the screen. Participants answered verbally before proceeding to the next question. After finishing all tasks, participants were asked to fill out standardized questionnaires, including the System Usability Scale (SUS) [4], the NASA Raw Task Load Index (TLX) [16] for workload assessment, and the User Engagement Scale Short Form (UES-SF) [32]. They also completed the Simulator Sickness Questionnaire (SSQ) [19] before and after the VR system usage to assess any changes in physical condition caused by it.

*6.1.5 Implementation.* The two systems are implemented on a Windows computer with an Intel i7 processor at 2.50 GHz and 16GB RAM. The VR system is built on Unity 3D with Oculus Quest SDK [31] with a smooth frame rate that exceeds 80 FPS. The desktop system (Figure 5) utilized ECharts [12] to build interactive Sankey diagrams, using the same color palette and having the same features as the VR system, including zooming, moving, highlighting, dragging, hovering labels, and reset. The desktop system runs on the Chrome browser with a 24-inch monitor at 2560x1440p resolution.

## 7 RESULTS

We report quantitative and qualitative results from the above study. We employed the same analysis method as the first user study.

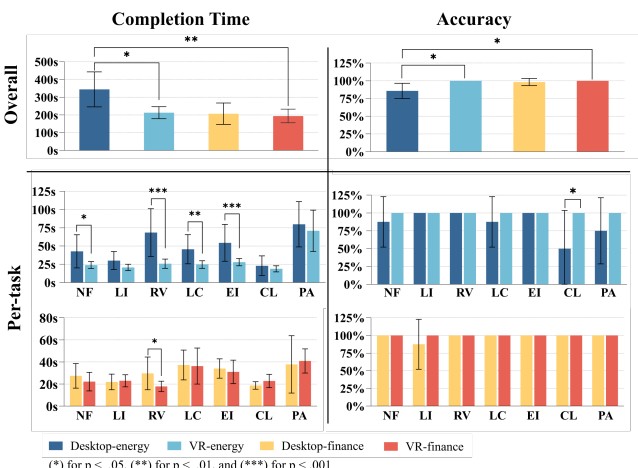

**Figure 6: Averages and standard deviations for task completion times and accuracy rates.**

### 7.1 Task Performance

We adopted the completion time and task accuracy to evaluate the task performance. Figure 6 presents averages and standard deviations for completion time, success rates, and overall and per task in the study. Overall, participants had a significantly better comprehension of the energy flow dataset in the VR system,

*7.1.1 Completion time.* Both the datasets (p < .01) and the systems (p < .01) had significant impacts on the task completion time with a strong interaction between them (p < .05). To further verify the effect of the Sankey diagram systems, we analyzed the results of each dataset separately.

In the case of the financial dataset, participants had a slightly quicker interpretation using the VR system, showcased by the less average overall time (p > .05). They found the VR system especially effective in the RV task compared to the desktop system, with significantly shorter completion time (p < .05). In the case of the energy flow dataset, participants spent a significantly shorter time comprehending the data flows in the VR system than the desktop system, demonstrated by less average overall completion time (p < .05) and less completion time of the NF (p < .05), RV (p < .001), LC (p < .01), and EI (p < .01) tasks.

*7.1.2 Task accuracy.* We used the Wilcoxon rank-sum test for accuracy analysis. Within the context of the financial dataset, participants interpreted the diagram accurately in both the VR and desktop systems. In the energy flow dataset scenarios, participants had a more precise understanding of the data flows using the VR system than the desktop system, demonstrated by significantly higher accuracy rates in the overall and CL tasks (p < .05).

In summary, using the VR system resulted in significantly better task performance in the energy dataset and slightly better performance in the financial dataset. We attribute the variation in performance across different datasets primarily to the complexity of Sankey diagrams in each dataset. Specifically, the energy dataset's Sankey diagram encompasses six layers, presenting a more intricate structure than the finance dataset's four-layer diagram. This

complexity likely required the advanced visualization capabilities of the VR system to ensure accurate and efficient comprehension.

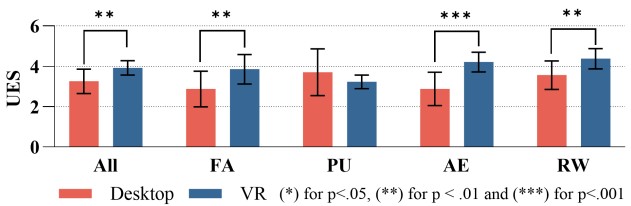

**Figure 7: The results of UES-SF questionnaire under the desktop and VR .**

## 7.2 User Engagement

We collected and analyzed the User Engagement Scale - Short Form (UES-SF) questionnaires regarding user engagement. The average scores for each component and the overall scores are presented in Figure 7. Participants felt more engaged in the VR system, demonstrated by the significantly higher average overall score of UES-SF and higher scores in Focused Attention (FA, p < .01), Aesthetic Appeal (AE, p < .001), and Reward Factor (RW, p < .01) sub-components. The average score of Perceived Usability (PU) was rated slightly higher with the desktop system, which may be due to the inherent familiarity with mouse manipulation.

We believe the immersive nature of VR technology contributes to the increase in engagement, which envelops users in a three-dimensional data landscape that demands and captures their full attention. In addition, the specific design elements, including the 3D form and the spatial interactions that we incorporated, are also relevant factors for 3D data exploration.

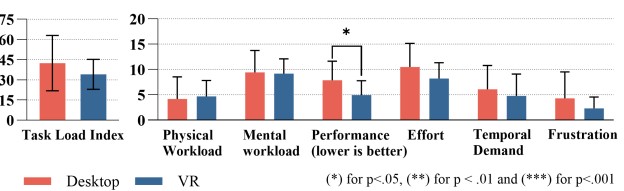

**Figure 8: Task workload components of the Nasa TLX questionnaire in the desktop and VR systems.**

## 7.3 Usability, Workload & Simulator Sickness

*7.3.1 Usability.* In general, both VR and desktop systems are rated "good" in usability. Participants found the VR system's usability slightly better than the desktop system, showcased by slightly higher scores of SUS (p > .05). This encouraging outcome suggests users' general acceptance and approval of our VR system. Lower usability scores were mainly due to the users' lack of familiarity with the VR controllers. They found it difficult to remember a variety of button functions. However, the majority of participants were able to quickly learn how to use the system, typically within 5-10

minutes of teaching and familiarization, and found the interactions smooth and intuitive. This positive response is noteworthy because most participants had little or no experience with VR technology.

*7.3.2 Workload.* As measured by the NTLX questionnaire (Figure 9), we found no significant difference in the overall workload between the VR and desktop systems. Considering the sub-components, participants felt more effective and successful in their performance using the VR system, reflected by the lower score (lower is better) in the performance sub-component (p < .05). This indicates that the VR system enhances participants' perceived performance without adding workloads.

*7.3.3 Simulator Sickness.* We found no significant difference between pre and post-exposure simulator Sickness scores (p > .05) with an average delta of 2.1 (SD = 12.95), which should be considered negligible. After spending an average of 18 minutes in VR, none of our participants reported experiencing any discomfort.

## 7.4 User Feedback

We present the feedback from our participants derived from semi-structured interviews.

**Most participants preferred the VR Sankey diagram system for task efficiency and entertainment value over the desktop system.** Out of the 16 participants, 12 found the VR system to be very helpful in completing tasks, while the remaining four preferred the desktop system (P5, P6, P11, P13). They mentioned that the choice between the two systems depends on the situation. The desktop system is more efficient in dealing with less complex data, whereas the VR system is more effective in handling denser data, as users can step into the diagrams and avoid obstruction by adjusting views. Two participants with extensive gaming and controller experience strongly preferred completing tasks with the VR system (P4 and P15). All participants highly praised VR for its entertainment value, using terms like "*more fun*" and "*more encouraging*". P10 commented, "*Viewing these three-dimensional diagrams in a VR environment makes me feel like the data is flowing, very vivid.*" In contrast, a couple of users mentioned "*the desktop experience is pretty dull for the given tasks.*"

**The form-switching feature was considered practical, with 3D form helping to inspect fine details and 2D form for the big picture.** Participants noted the form-switching feature improves the flexibility of the VR system. The 3D form enables users to walk into the diagram space, making it easier to observe and interact with tiny links (P5, P6, P8, P12). As for the 2D form, participants confirmed its straightforward view of diagrams, which improved their understanding of the data's hierarchies (P7-10, P15). P15 described the form switching as "*a clever feature that enables me to handle various data visualization challenges more effectively.*"

**Spatial highlighting was considered a more effective way to distinguish the selected data flows from others.** Some participants favored the highlighting feature in our VR system, which brings highlighted information to the front and makes it stand out from the background without obstructing the view of non-highlighted information (P10, P12, P14-16). They found it particularly helpful for the PA task because they could easily extract

information from the diagram and keep searching in the background. However, the highlighting feature in the desktop system, without spatial depth, can only enhance color contrast, making it hard to focus on the desired information.

**The dragging feature in VR provided users with a sense of control and enjoyment.** Participants like the dragging feature in the VR system, even if they tend to use it more frequently in the desktop system. Dragging node operation is not essential for task completion in VR but enhances concentration and enjoyment while organizing or grouping data. As P8 said, "*The experience of dragging in the VR system is enjoyable and smooth. I can easily move the nodes to any desired position, which gives me a sense of control over the data.*"

## 8 DISCUSSIONS

This section discusses the design lessons we learned, the limitations of this study, and the potential future work.

### 8.1 Design Lessons

**Sankey diagrams in VR incorporating VR's additional spatial depth dimension.** In this paper, we proposed a 3D Sankey diagram design incorporating the VR's additional spatial depth dimension. Specifically, we increased the thickness of each node and rearranged the nodes within each layer to utilize the depth dimension, thereby clarifying the relationships between data points and the trends of links. Most participants appreciated that these modifications in the 3D Sankey diagram design were beneficial for a better understanding and an engaging experience of analysis and exploration of complex data flows within the diagrams. However, participants also expressed that the 3D forms look more complex in simple datasets, making them even more challenging to understand. The 2D Sankey diagrams were ideal for simple datasets since participants stated that the 2D forms offer a quick overview to help them find the nodes quickly. These observations highlight that 3D Sankey diagrams may not always be appropriate for all cases, and 2D and 3D forms have their own advantages in certain situations. Thus, future research on incorporating additional depth dimensions into visualizations should first consider whether it is suitable for 3D adaptation and whether the complexity of the data has reached a level that necessitates the third dimension to convey more information. Moreover, future studies might also consider supporting users to seamlessly switch between 2D and 3D forms, which would combine the advantages of both forms and empower users to select the most fitting forms for their specific needs and preferences.

**Interactive features of the visualization systems in VR leveraging spatial capacity.** We designed highlighting and dragging features with spatial capacity in our VR system. Specifically, spatial highlighting enhances the color and emphasizes the elements by setting their position in front of the diagram, thereby making the highlighted elements stand out from the others without obstructing them. The majority of participants praised this interaction, as they could quickly identify and focus on important information without being distracted by the surrounding data points, enhancing the analytical tasks' efficiency. The dragging feature allows participants to rearrange the diagrams by dragging

the nodes to wherever they want without breaking the connections between them. The participants noted that this feature was unnecessary for task completion but gave them a sense of control. Dragging the nodes in VR using controllers made them feel like they were grasping them with actual hands, with a feeling of full-body involvement in the interaction. It underscores the importance of designing visualization interactions combining spatial characteristics in VR environments for more efficient analytical activities and a more engaged experience.

### 8.2 Limitations & Future Work

The visualization of the 3D Sankey diagram should be improved. Some participants noted that using shadows to indicate hierarchical information lacks clarity and precision (P2, P15). We plan to place nodes of the same layer inside a box and add textual descriptions to denote their categories to clarify hierarchical information. Besides, the layout logic of the 3D Sankey diagram could be further optimized to reduce link overlaps with linear programming. Also, the interactions could be further explored to enhance immersive analytical activities. Some participants recommended adding more advanced functions, such as node filters, gestural manipulation of the nodes, or eye tracking for highlighting information (P15 and P16).

## 9 CONCLUSION

In conclusion, our research on developing, implementing, and evaluating the immersive Sankey diagram system in VR has provided valuable insights and contributions to the area of immersive analytics. We created a VR Sankey diagram system to enhance complex data flow visualization and interpretation. The user studies further confirmed its advantages of improved task performance, engagement, and reduced workload over the traditional desktop-based system, especially when presenting intricate data flows. Our work highlights the significant potential of VR in data visualization and attempts to inspire further advancements in designing and developing VR-based visualization systems.

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
