# OpenReview forum: "Designing Spatial Visualization and Interactions of Immersive Sankey Diagram in Virtual Reality"
_acmmm.org/ACMMM/2024/Conference — MM2024 Oral_

### Official Review · Reviewer_pGfH · 2024-05-22

**Rating:** 5
**Confidence:** 4

**Summary:**

This article develops, implements and evaluates an immersive Sankey diagram system in a VR environment to enhance the visualization and interpretation of complex data streams. The authors conducted two user studies, adapted the color settings and interaction features based on the results of the first user study, and conducted a subsequent second user experiment. The results of the user experiments confirm the benefits of the immersive Sankey diagram system in terms of improved task performance, engagement, and reduced workload, especially when presenting complex data streams. Valuable insights and contributions to the field of immersive analytics are provided.

**Strengths:**

This paper presents an innovative 3D Sankey diagram design that utilizes the spatial and immersive properties of virtual reality to significantly differentiate it from the traditional 2D Sankey diagram. The paper provides solid empirical evidence through two well-structured user studies that evaluate in detail the effectiveness of the 2D and 3D Sankey diagram designs, as well as a comparison of the VR system with a traditional desktop system. The authors describe the design of the evaluation task in detail from a theoretical and well-reasoned perspective.

Overall, I found the article to be logical and easy to read, with a reasonable experimental design and a detailed analysis of the results.

**Limitations:**

The general content of the paper is reasonable to me, and there are some detailed parts of the paper that may need to be double-checked or revised by the author.

1. I noticed that in both user studies, the authors used a number of standardized questionnaires for evaluation. For example, SEQ, SUS, NASA-TLX, UES-SF, and SSQ, but the details of these scales are missing in the article. I think that the choice of these scales is appropriate, but I suggest that the authors could provide a short introduction to these scales in the appropriate parts of Section 4 and Section 6 so that readers who are not familiar with these scales can quickly understand the approximate assessment use and content of the scales.

2. In the two user studies, participants were required to complete five tasks and seven tasks, respectively. However, there are no links or tables in the text to tell the reader what questions these specific tasks required participants to answer. The authors can probably provide an online link or an appendix to add to the question content. It would be easier for the reader to understand.

3. section 7.3.2, line 760: I didn't find Figure 9, I think the author was referring to Figure 8. And I'm confused about the NTLX acronym, it seems like the NASA-TLX acronym is the more common form.

4. For Figure 2 and Figure 6, I have a small question about why there is a need to split 2D-low, 3D-low and 2D-high and 3D-high into top and bottom graphs. This seems to add to the reading difficulties. Why not put it in one diagram like Figure 3, and the same for Figure 6?

**Suitability:**

3

---

### Official Review · Reviewer_XmW9 · 2024-05-23

**Rating:** 4
**Confidence:** 3

**Summary:**

This article presents a study where a virtual reality (VR) environment is used to create and assess an immersive Sankey diagram system. The goal is to improve the visualisation and understanding of intricate data streams. The authors performed two user studies, modified the colour settings and interaction elements based on the findings of the first user research, and then carried out a second user trial. The findings from the user trials validate the advantages of the immersive Sankey diagram system in terms of enhanced task performance, engagement, and decreased burden, particularly when displaying intricate data streams. Significant and valuable insights and contributions are offered in the area of immersive analytics.

**Strengths:**

Regarding the merits of the paper, this study introduces a novel design for a 3D Sankey diagram that makes use of the spatial and immersive features of virtual reality to distinguish it from the conventional 2D Sankey diagram. The report presents robust empirical data from two meticulously designed user tests that thoroughly assess the efficacy of 2D and 3D Sankey diagram designs. Additionally, it includes a comparison between a virtual reality (VR) system and a conventional desktop system. The authors provide a comprehensive and logical description of the design of the assessment assignment, drawing on theoretical principles and sound reasoning.

**Limitations:**

As for the weakness of the paper, a number of questionnaires have been employed, but they are clearly explained and justified. Another question I wonder is when the reading of the 3D Sankey diagram become extremely difficult. Are there any extreme cases (both good or bad). Other minor issues include typo and misreference (e.g., Figure 9 -> Figure 8 in l760).

**Suitability:**

2

---

### Official Review · Reviewer_P7Lm · 2024-05-23

**Rating:** 4
**Confidence:** 2

**Summary:**

This paper explores the use of virtual reality (VR) to enhance the visualization and interactivity of Sankey diagrams, which are commonly used to represent data flows. The main contributions of the paper include the development of a new 3D Sankey diagram system in VR, demonstrating its advantages through user studies, and providing design insights for future immersive visualization tools. The study highlights the significant potential of VR in enhancing data visualization, particularly for complex data flows represented by Sankey diagrams. The findings indicate that immersive and interactive VR environments can offer a better analytical experience and greater user engagement compared to traditional 2D visualizations.

**Strengths:**

The paper introduces a novel approach by applying VR technology to Sankey diagrams, which is relatively unexplored. This approach leverages the immersive and spatial capabilities of VR to enhance data visualization, offering a fresh perspective on how complex data flows can be represented and interacted with. The following is some key information:

1. The situates its contribution within the framework of immersive analytics, a burgeoning field that aims to integrate data visualization with immersive user interfaces. This theoretical grounding provides a sturdy foundation for the research and highlights the potential benefits of VR in data analysis. Specifically, the related work section provides examples of the interaction and application of various statistical charts.

 2. This paper includes two comprehensive user studies comparing the VR system with traditional desktop-based systems. These studies are well-designed, with clear tasks and metrics for evaluation. The results provide robust evidence of the advantages of the VR system in terms of task performance, user engagement, and reduced cognitive workload.

**Limitations:**

The paper lacks in-depth technical details on how the VR system was implemented, specifically regarding the algorithms used for the 3D layout and interaction techniques. The following questions which can help improve this article:

1. The lack of extensive comparison with other visualization methods leaves some uncertainty about how much the VR approach truly advances the field. References to works like Millais et al. (2018) and Wagner Filho et al. (2019) show that there are existing VR visualization methods that could serve as useful benchmarks.

2. More comprehensive technical descriptions would aid in understanding the innovations and could be beneficial for researchers looking to replicate or build upon this work. How do you get users to make sankitu in two different devices and environments? How do you make interactive systems to put Sankey diagrams in? I think these can be explained in detail in the paper.

3. While visual clarity is important, the primary goal of data visualization is to aid in analysis and decision-making. More emphasis on these aspects would strengthen the paper's claims about the system's utility.

4. User studies included only a small number of participants (7 in the first study and 16 in the second). A larger and more diverse sample size would enhance the generalizability of the findings and provide more robust evidence of the system's effectiveness.

5. The original text states and mistake: "We found no significant difference between pre and post-exposure simulator Sickness scores (p > .05) with an average delta of 2.1 (SD = 12.95), which should be considered negligible." It can be revised to: "We found no significant difference between pre- and post-exposure simulator sickness scores (p > .05), with an average delta of 2.1 (SD = 12.95), which should be considered negligible." This revision provides a more precise description of your statistical data.

**Suitability:**

1

---

### Official Review · Reviewer_ytUW · 2024-05-26

**Rating:** 5
**Confidence:** 3

**Summary:**

This paper presents an immersive implementation of Sankey diagrams which makes use of the 3D space and interaction techniques to enhance data flow visualisation. The authors also present the results of two user studies, comparing their 3D solution to a standard 2D presentation of Sankey diagrams in a VR environment, as well as comparing an interactive VR application to an interactive desktop application.

**Strengths:**

The paper is well-written and the presented solution (i.e., the immersive, interactive, three-dimensional presentation of Sankey diagrams) appears to be well designed both in regard to the utilization of 3D space and interaction design. The methodology design for the two user studies is sound and well thought out, and the supplementary material provides additional context to readers. The results of user studies certainly seem promising, supporting the presented solution while also offering design lessons for the future.

**Limitations:**

My primary concern with this paper is the very small sample size of participants used to evaluate the presented solutions, particularly in the first study (N = 7). Given the study design, which does not appear to be overly demanding for either participants or administrators, I am curious about the rationale behind such a small sample. This limitation likely impacts the variability of the results and reduces their statistical power, making it difficult to draw strong conclusions.

Related to the sample size issue, the Limitations & Future Work section focuses on potential improvements to the presentation of the Sankey diagram (which is the main contribution to the paper), however it does not adequately address the limitations and potential improvements in participant recruitment and study design.

While the supplementary material includes the set of questions asked during the experiments, the paper could benefit from additional context or examples within the main text. For instance, the sentence "Specifically, we represent the tasks with a set of questions that participants need to answer to evaluate the task performance using the systems" would be clearer with added context or a note directing readers to the supplementary material. Perhaps the presentation of questions in the supplementary material could also be improved if their connection to the corresponding task was also explicitly stated.

I feel that placing Figure 1 closer to its mention in section 4.1 could improve the readability of the paper.

**Suitability:**

3

---

### Meta-Review · Area_Chair_PbSo · 2024-07-09

**Recommendation:** Accept (Oral)
**Confidence:** 4

**Metareview:**

The paper presents the design, development and evaluation of an an immersive Sankey diagram system in a VR environment to enhance the visualization and interpretation of complex data streams. The approach makes sense and is valuable but there are limitations in terms of sample size.

Thanks to the authors for their efforts and for the rebuttal.

Please ensure all concerns are addressed for the final version (with exception of sample size) all of which are quite doable.

A shepherd might be useful with this paper.